   

# Mre11 exonuclease activity promotes irreversible mitotic progression under replication stress

Yoshitami Hashimoto ⬤, Hirofumi Tanaka

**Mre11 is a versatile exo-/endonuclease involved in multiple aspects of DNA replication and repair, such as DSB end processing and checkpoint activation. We previously demonstrated that forced mitotic entry drives replisome disassembly at stalled replication forks in *Xenopus* egg extracts. Here, we examined the effects of various chemical inhibitors using this system and discovered a novel role of Mre11 exonuclease activity in promoting mitotic entry under replication stress. Mre11 activity was necessary for the initial progression of mitotic entry in the presence of stalled forks but unnecessary in the absence of stalled forks or after mitotic entry. In the absence of Mre11 activity, mitotic CDK was inactivated by Wee1/Myt1–dependent phosphorylation, causing mitotic exit. An inhibitor of Wee1/Myt1 or a nonphosphorylatable CDK1 mutant was able to partially bypass the requirement of Mre11 for mitotic entry. These results suggest that Mre11 exonuclease activity facilitates the processing of stalled replication forks upon mitotic entry, which attenuates the inhibitory pathways of mitotic CDK activation, leading to irreversible mitotic progression and replisome disassembly.**

## Introduction

Eukaryotic DNA replication is initiated at replication origins during the S-phase of the cell cycle, forming replication forks to which the replisome, a multi-protein complex, binds and conducts DNA replication. The replisome contains the Cdc45/MCM2–7/GINS helicase complex and the replicative DNA polymerases Pol$\alpha/\delta/\varepsilon$, as well as many other accessory factors with specialized functions (Burgers & Kunkel, 2017). When two progressing forks converge, DNA replication is locally terminated, the daughter strands are decatenated by topoisomerase II, and the replisomes are disassembled by p97 ATPase that recognizes MCM7 poly-ubiquitylated by cullin E3 ligase (CRL2$^{Lrr1}$ in metazoa) (Maric et al, 2014; Moreno et al, 2014; Dewar et al, 2017; Sonneville et al, 2017). However, replication forks can often encounter obstacles for their progression, such as DNA lesions, nucleotide reduction, secondary DNA structures, DNA–RNA

hybrids, and transcriptional machinery (Zeman & Cimprich, 2014; Gaillard et al, 2015). These replication stress factors cause stalled replication forks, which are stabilized for replication restart until the stress is relieved. Recently, increasing evidence suggests that stalled forks are regularly remodeled into reversed forks in which the two parental strands reanneal, and the two nascent strands also anneal to form a chicken footlike four-way junction (fork reversal) (Neelsen & Lopes, 2015; Poole & Cortez, 2017). Reversed forks are believed to contribute to fork stabilization and repair, and stalled forks (or reversed forks) have been shown to activate ATR-Chk1 checkpoint signaling to down-regulate CDK1/2 activities via Cdc25 inactivation, suppressing de novo initiation of DNA replication and cell cycle progression into G2/M phases (Saldivar et al, 2017). Without replication stress, Cdc25 usually activates Cdk1/2 during S to G2/M phase transition to drive replication initiation and cell cycle progression by removing Wee1/Myt1 kinase–mediated inhibitory phosphorylation of Thr14 and Tyr15 (Sørensen & Syljuåsen, 2012; Crncec & Hochegger, 2019; Elbæk et al, 2020).

Mre11, a versatile nuclease with both exo- and endonuclease activities, functions within the Mre11–Rad50–Nbs1 (MRN) complex that is involved in many aspects of DNA replication and repair, such as DSB end resection for homologous recombination-dependent DNA repair, ATM activation in response to DSBs, and DNA tethering between sister chromatids (Paull, 2018; Syed & Tainer, 2018; Reginato & Cejka, 2020). In addition to these physiological roles, pathological fork resection by Mre11 has also been reported, revealing its role in a fork protection mechanism that involves BRCA2, Rad51, and other DNA damage response proteins (Pasero & Vindigni, 2017; Rickman & Smogorzewska, 2019). A reversed fork comprises a regressed arm with two nascent DNA strands of different lengths, exposing a single-stranded DNA (ssDNA) region. BRCA2 promotes the assembly of filament Rad51 onto this ssDNA region to stabilize the reversed fork under replication stress. On the contrary, in BRCA2-deficient cells, this unprotected ssDNA region serves as an entry point for Mre11 to bring about uncontrolled fork degradation (Schlacher et al, 2011; Kolinjivadi et al, 2017; Taglialatela et al, 2017).

The fork stabilization system ensures that replisomes are stably maintained at stalled forks (or maybe reversed forks) under replication stress, which enables replication restart without de novo

---

School of Life Sciences, Tokyo University of Pharmacy and Life Sciences, Hachioji, Japan

Correspondence: hashimo@toyaku.ac.jp

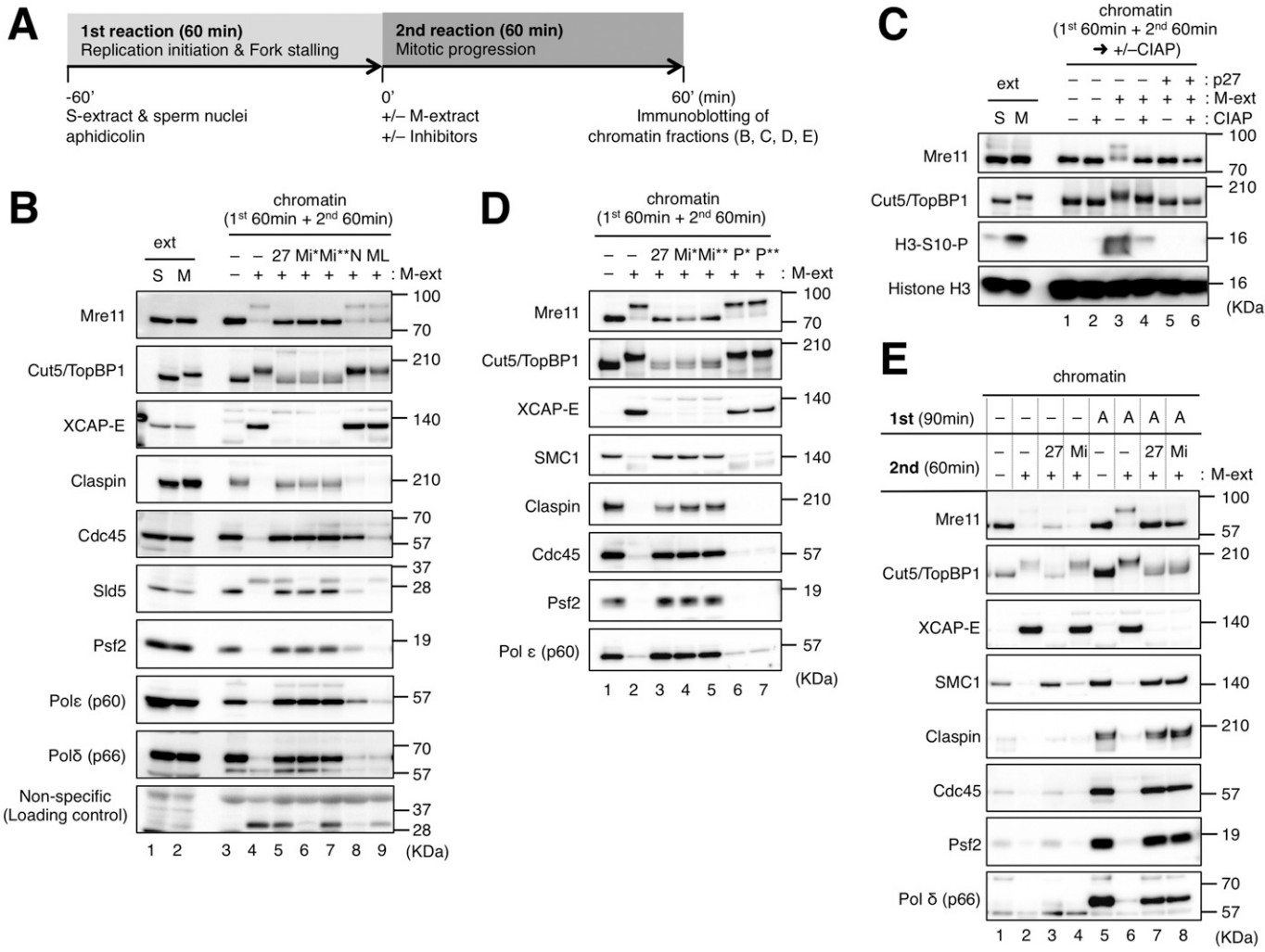

**Figure 1. Mre11 exonuclease activity is required for mitotic entry and mitotic replisome disassembly in the presence of stalled replication forks.**
**(A)** The basic experimental design. In the first reaction, sperm nuclei (5,000/µl) were incubated for 60 min in 20 µl of S-phase/interphase egg extract (S-extract) with 10 µg/ml of aphidicolin. In the second reaction, 20 µl of M-phase egg extract (M-extract) was added to the first reaction mixture together with an additional 10 µg/ml of aphidicolin to induce mitotic entry and replisome disassembly. After 60-min incubation of the second reaction, chromatin fractions were isolated and analyzed by immunoblotting. **(B, D)** The effects of inhibitors on mitotic entry and replisome disassembly. The second reaction was performed in the absence or presence of 100 µg/ml of His-p27 (27), 50 µM mirin (Mi*), 100 µM mirin (Mi**), 100 µM NMS-873 (N), 10 µM MLN-4924 (ML), 50 µM PFM01 (P*), or 100 µM PFM01 (P**). **(C)** Role of phosphorylation in mitotic mobility shifts of Mre11 and Cut5/TopBP1. The isolated chromatin fractions were treated with or without calf intestine alkaline phosphatase (+/−CIAP) and analyzed by immunoblotting. In (B) and (C), 0.5 µl of S-phase and M-phase extracts (ext S, M) was analyzed by immunoblotting as controls. **(E)** Comparison of mitotic entry between replication-complete and replication-incomplete nuclei. The first reaction was performed for 90 min with (replication-incomplete) and without (replication-complete) aphidicolin.
Source data are available for this figure.

assembly of replisomes in the S-phase (Bellelli & Boulton, 2021; Guilliam, 2021). However, these stabilized replisomes are disassembled by increasing mitotic CDK activity that leads to cell cycle progression into mitosis (Hashimoto & Tanaka, 2018). The RING E3 ligase TRAIP mediates the poly-ubiquitylation of MCM7, which promotes p97-dependent replisome disassembly in the M-phase (Deng et al, 2019; Moreno et al, 2019; Sonneville et al, 2019). Exploiting *Xenopus* egg extracts with interphase and mitotic activities, we previously established an in vitro system that reconstitutes mitotic replisome disassembly (MRD) under replication stress (Hashimoto & Tanaka, 2018). In this study, we further investigated the molecular mechanism of MRD using this system and chemical inhibitors. Unexpectedly, we found that Mre11 exonuclease activity is required for

mitotic entry under replication stress, possibly through the processing of stalled or reversed forks, which in turn promotes MRD.

## Results

### Mre11 exonuclease activity is required for mitotic entry and replisome disassembly in the presence of stalled replication forks

We previously demonstrated that mitotic entry drives replisome disassembly at stalled replication forks in *Xenopus* egg extract (Hashimoto & Tanaka, 2018) (Fig 1A). In this system, sperm nuclei are

first incubated in S-phase (interphase) egg extract (S-extract) in the presence of aphidicolin, an inhibitor of replicative DNA polymerases $\alpha/\delta/\varepsilon$, which allows initiation but not elongation and causes all replication forks to be stalled. An equal volume of M-phase extract (M-extract) is then supplemented to induce mitotic entry and replisome disassembly. MRD has been shown to be mechanistically different from interphase replisome disassembly, which occurs during replication termination or replication fork collapse (Wu et al, 2021).

We first tested various inhibitors for their effects on mitotic entry and MRD by the immunoblotting of the chromatin fractions (Fig 1B). Major replisome components such as claspin, GINS subunits (Sld5 and Psf2), and Pol$\delta/\varepsilon$ were stably maintained on chromatin until the end of the second incubation without the induction of mitotic entry, whereas they largely dissociated from chromatin during the second incubation with induction of mitotic entry (Fig 1B, lanes 3 and 4), indicating MRD had occurred. Mitotic entry was confirmed by chromatin binding of XCAP-E, a common subunit of condensin I/II. The mobility shifts of Mre11 and Cut5/TopBP1 were also observed upon mitotic entry. Mre11 reverted to its original state after treating chromatin with alkaline phosphatase (Fig 1C), demonstrating that the mobility shift of Mre11 on chromatin was exclusively due to mitotic phosphorylation. The mobility of Cut5/TopBP1 on chromatin did not fully revert to its interphase state but shifted closer to its cytosolic state in M-extract after phosphatase treatment (Fig 1C), suggesting a partial role of phosphorylation in the mobility shift. It is also possible that mitotic Cut5/TopBP1 undergoes additional modification other than phosphorylation. Inactivation of mitotic CDK activity with recombinant p27 nullified the effect of the addition of M-extract for mitotic entry and MRD (Fig 1B, lane 5; Fig 1C lanes 5 and 6). NMS873 (an inhibitor of p97 ATPase) partly inhibited MRD but did not inhibit mitotic entry (Fig 1B, lane 8). MLN4924 (an inhibitor of cullin Ub ligases) did not suppress mitotic entry or MRD (Fig 1B, lane 9). These results were in accordance with our previous results (Hashimoto & Tanaka, 2018). Intriguingly, we found that mitotic entry and MRD were both suppressed by the presence of mirin, an inhibitor of the exonuclease activity of Mre11 (Fig 1B, lanes 6 and 7) (Dupré et al, 2008). In contrast, PFM01, an inhibitor of the endonuclease activity of Mre11, did not have any inhibitory effect (Fig 1D) (Moiani et al, 2018). We then examined whether the effect of mirin is restricted to situations of replication stress. When the first incubation was extended to 90 min and performed without aphidicolin to allow for the completion of DNA replication, mirin did not inhibit mitotic entry, and a small amount of residual replisome components were dissociated from chromatin (Fig 1E, lanes 1, 2, and 4). In the presence of aphidicolin, mirin inhibited mitotic entry and MRD even after the longer first incubation (Fig 1E, lanes 5, 6, and 8). By contrast, recombinant p27 inhibited mitotic entry and MRD irrespective of the presence of aphidicolin (Fig 1E, lanes 3 and 7). In addition, mirin did not inhibit interphase replisome disassembly in the absence of aphidicolin, as opposed to NMS873 which caused persistent chromatin binding of claspin, Psf2, and poly-ubiquitylated Mcm7 at later time points (Fig 2A). These results suggest that the exonuclease activity of Mre11 is required for mitotic entry and MRD in the presence of stalled replication forks but not when DNA replication is nearly complete and without exogenous inhibition.

## Mre11 facilitates the processing of stalled forks during the early stages of mitotic entry

As mirin was used together with M-extract in our experiments described so far, Mre11 could have the potential to act on stalled forks during interphase before mitotic induction. However, our results show that such activity, if any, was not sufficient to promote mitotic entry. Alternatively, Mre11 activity may be suppressed during interphase, possibly through a fork protection mechanism (Pasero & Vindigni, 2017; Rickman & Smogorzewska, 2019). To clarify the timing at which Mre11 activity is required, we added mirin at different time points on and after mitotic induction and monitored chromatin association of replisome-related factors. When mirin was added at 15 or 30 min after mitotic induction, the appearance of the mitotic forms of Mre11 and Cut5/TopBP1 were not affected, and XCAP-E accumulated on, and the replisome components claspin/Cdc45/Ps2/Pol$\delta$ disappeared from chromatin over time in a similar fashion to the case without mirin (Fig 2B, lanes 1–3 and 7–12), indicating that mirin had no inhibitory effect after the initiation of mitotic entry. When added at 0 min, mirin completely inhibited mitotic entry and MRD (Fig 2B, lanes 4–6). These results suggest that Mre11 acts on stalled forks during an initial 15-min period after mitotic induction and that Mre11 activity is required for subsequent mitotic progression.

We then examined whether interphase Mre11 activity is also involved in mitotic entry and MRD (Fig 2C). To suppress Mre11 activity only in the interphase, we added mirin at 100 $\mu M$ in the first incubation and diluted it by one-fifth (to 20 $\mu M$) with 1.5 volumes of S-extract and 2.5 volumes of M-extract in the second incubation. In this condition, mirin hardly inhibited mitotic entry and MRD as with the case that 20 $\mu M$ of mirin was added only in the second incubation (Fig 2C, lanes 3 and 6). When 100 $\mu M$ of mirin was added in the second incubation, mitotic entry and MRD were inhibited irrespective of the presence or absence of mirin in the first incubation (Fig 2C, lanes 4 and 7). These results suggest that interphase Mre11 activity is not required for mitotic entry and MRD. On the other hand, the addition of recombinant p27 at 15 or 30 min stopped mitotic progression as indicated by the unloading of XCAP-E and the mobility shift of Mre11 to an interphase form (Fig 2D, lanes 7–12). MRD was also inhibited by p27 when added at 15 min but not at 30 min. These results show that Mre11-dependent processing of stalled forks during the early stages of mitotic entry is necessary and sufficient for mitotic progression, whereas mitotic CDK activity is continuously required for mitotic progression.

## Mitotic CDK activity is inactivated without Mre11-dependent processing of stalled forks

We then assessed whether the inhibition of Mre11 activity would affect the overall activity of mitotic CDK by the immunoblotting of whole extracts containing both nuclei and cytosol (Fig 3A and B). APC3 is a subunit of the anaphase-promoting complex/cyclosome E3 ubiquitin ligase responsible for cyclin B degradation (Fujimitsu et al, 2016), and Mcm4 is a subunit of the minichromosome maintenance complex MCM2-7 that constitutes the replicative helicase Cdc45/MCM2-7/GINS complex together with Cdc45 and GINS at the replication fork (Burgers & Kunkel, 2017). Because both APC3 and Mcm4 are highly phosphorylated by mitotic CDK in mitosis

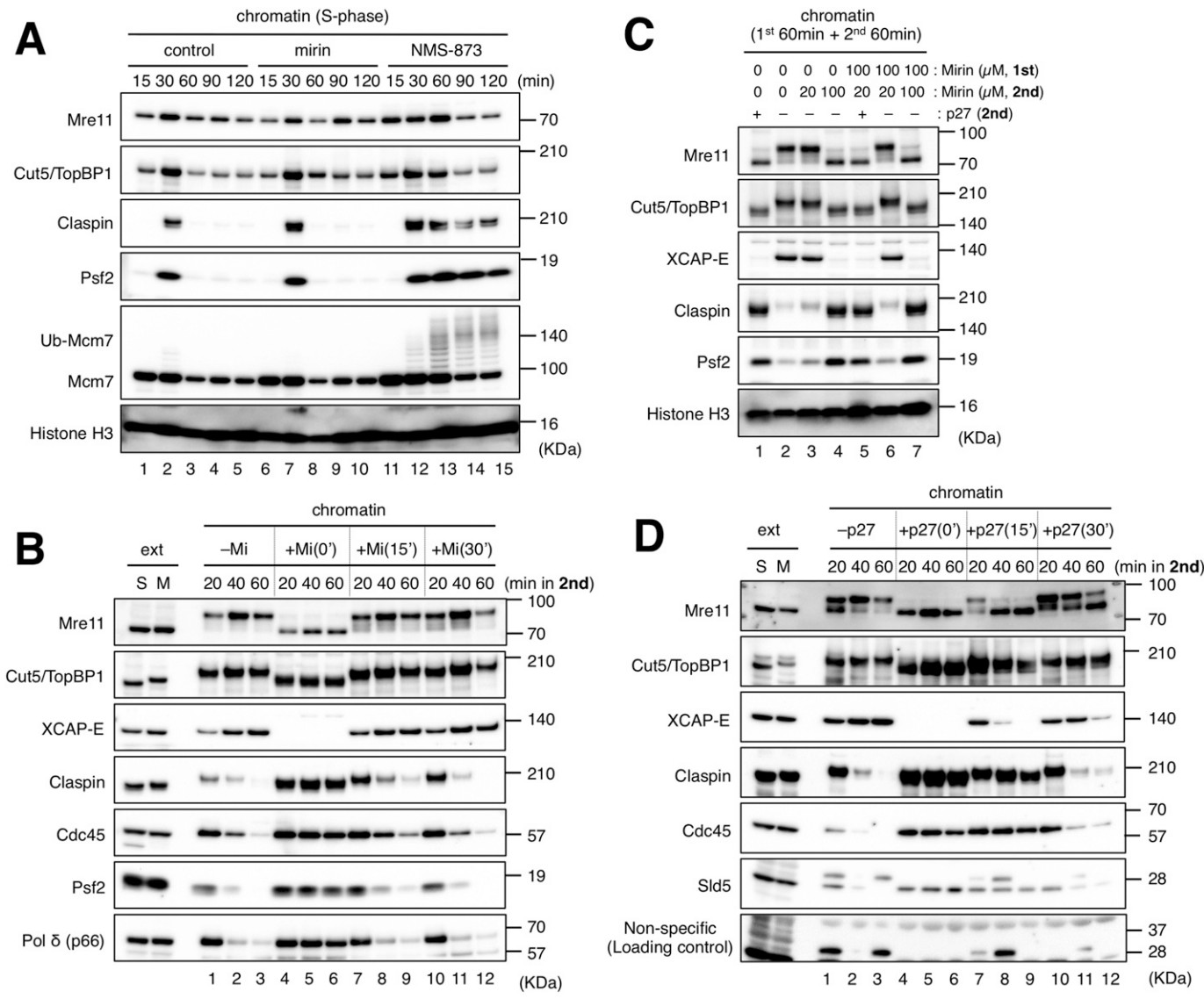

**Figure 2. Mre11 facilitates the processing of stalled forks during the early stages of mitotic entry.**
**(A)** The requirement of Mre11 activity for replisome disassembly during replication termination in the S-phase. Sperm nuclei (5,000/μl) were incubated for the indicated times in 20 μl of S-extract in the absence or presence of 100 μM mirin or 100 μM NMS-873. The chromatin fractions were isolated and analyzed by immunoblotting. **(B)** The requirement of Mre11 activity for mitotic progression after mitotic entry. A similar experiment to that shown in Fig 1B was performed under the following conditions: 100 μM mirin was added together with M-extract (+Mi(0′)) or at 15 (+Mi(15′)) and 30 min (+Mi(30′)) after the addition of M-extract. After 20, 40, or 60 min for the second reaction, the chromatin fractions were isolated and analyzed by immunoblotting. **(C)** The requirement of interphase Mre11 activity for mitotic entry. In the first reaction, sperm nuclei (5,000/μl) were incubated for 60 min in 20 μl of S-extract in the absence or presence of 100 μM mirin. In the second reaction, 30 μl of S-extract and 50 μl of M-extract were added (to dilute sperm nuclei to 1,000/μl) with or without additional mirin to adjust the final concentrations of mirin to 0, 20, and 100 μM as indicated. The concentrations of aphidicolin were kept at 10 μg/ml throughout the experiment in all the cases. His-p27 was added as controls. After 60-min incubation, the chromatin fractions were isolated and analyzed by immunoblotting. **(D)** The requirement of mitotic CDK activity for mitotic progression after mitotic entry. **(B)** The same experiment as shown in (B) was performed using His-p27 instead of mirin, and the chromatin fractions were isolated and analyzed by immunoblotting. In (B) and (D), 0.5 μl of S-phase and M-phase extracts (ext S and M) was also analyzed by immunoblotting as controls.
Source data are available for this figure.

but dephosphorylated in interphase, their phosphorylation states are good markers for overall mitotic CDK activity (Krasinska et al, 2011; Fujimitsu et al, 2016). Mitotic CDKs are mostly represented by cyclin B-CDK1 and cyclin A-CDK1, and their activities are negatively regulated by Wee1/Myt1–dependent phosphorylation of Thr14 and Tyr15 of CDK1 during the cell cycle (Sørensen & Syljuåsen, 2012; Crncec & Hochegger, 2019; Elbæk et al, 2020).

In the control reaction mixture of S-phase and M-phase extracts containing nuclei with stalled forks, both APC3 and MCM4 had two bands representing hyper-phosphorylated and hypo-phosphorylated forms (Fig 3A, lane 1, P-APC3/APC3 and P-MCM4/MCM4). The two APC3 bands remained constant for 60 min, whereas a most MCM4 turned into the hyper-phosphorylated form at 15 min and remained so until 60 min (Fig 3A, lanes 2–4). The inhibitory

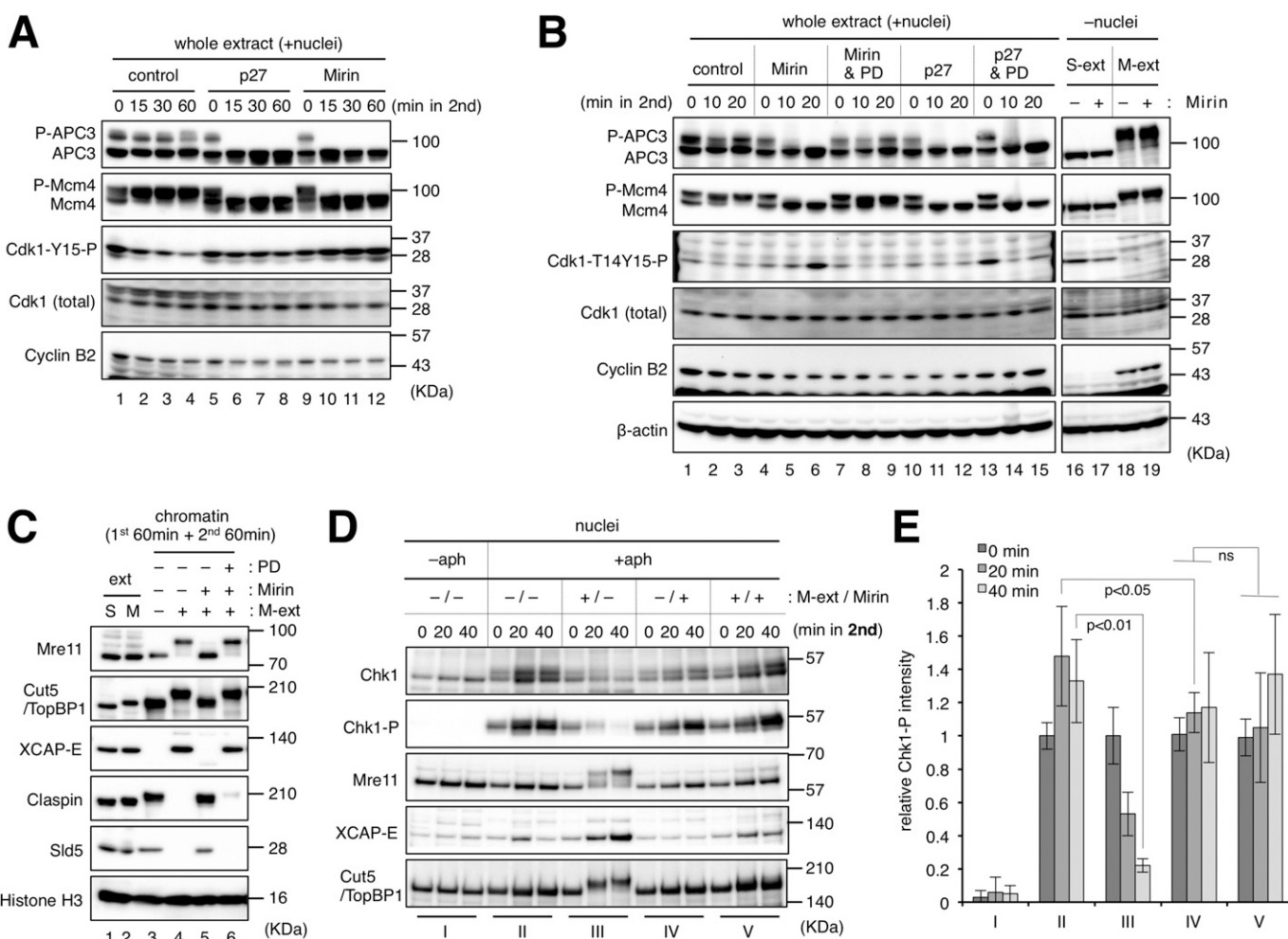

**Figure 3. Mitotic CDK activity is inactivated by Wee1/My1 in the absence of Mre11-dependent processing of stalled forks.**
**(A)** Inactivation of mitotic CDK activity in the presence of mirin. The same experiment with that shown in Fig 1B was performed with the indicated conditions. After 0-, 15-, 30-, and 60-min incubation of the second reaction, 1 μl of the whole extract was sampled and analyzed by immunoblotting. **(B)** The dependence of CDK inactivation on Wee1/Myt1 kinases. The same experiment with that shown in Fig 1B was performed with the indicated conditions. After 0-, 10-, and 20-min incubation of the second reaction, 1 μl of the whole extract was sampled and analyzed by immunoblotting (lanes 1–15). As controls, S-phase and M-phase extracts (S-ext and M-ext) were incubated without nuclei for 40 min in the absence and presence of mirin, and 1 μl of each were sampled and analyzed by immunoblotting. **(C)** The recovery of mitotic entry by the inhibition of Wee1/Myt1. The same experiment with that shown in Fig 1B was performed with the indicated conditions, and the chromatin fractions were isolated and analyzed by immunoblotting. **(A, B, C, D)** His-p27 (p27), mirin, and PD166285 (PD) were used at final concentrations of 100 μg/ml, 100 μM, and 10 μM, respectively. **(D, E)** The effect of mirin for the ATR-Chk1 checkpoint activity. The same experiment with that shown in Fig 1B was performed with the indicated conditions, and the nuclear fractions were isolated and analyzed by immunoblotting. The relative mean values of Chk1-P intensities of three independent experiments were shown in the graph using Cut5/TopBP1 intensities as loading controls. Error bar, ± S.D. *P*-values were calculated by the unpaired *t* test (one-tailed). ns, not significant. Source data are available for this figure.

phosphorylation of CDK1 at Tyr15 gradually decreased during the 60 min. By contrast, in the presence of p27, both APC3 and MCM4 became hypo-phosphorylated at 15 min and remained so until 60 min, and the basal level of Tyr15 phosphorylation of CDK1 was maintained during the 60 min (Fig 3A, lanes 5–8). Intriguingly, similar patterns of hypo-phosphorylation of APC3 and MCM4 were also observed in the presence of mirin (Fig 3A, lanes 9–12). Moreover, Tyr15 phosphorylation of CDK1 appeared to increase during the experiment. The total amounts of CDK1 and cyclin B2 were constant throughout the experiment in all the cases (Fig 3A, lanes 1–12). These results suggest that mitotic CDK activity is inactivated through the inhibitory phosphorylation at Tyr15 of CDK1 in the presence of mirin.

Because Wee1/Myt1 are the known kinases responsible for phosphorylating CDK1 at Thr14 and Tyr15 (Sørensen & Syljuåsen, 2012; Crncec & Hochegger, 2019; Elbæk et al, 2020), we tested whether an inhibitor for Wee1/Myt1, PD166285 can rescue the inactivation of mitotic CDK in the presence of mirin (Fig 3B). In the presence of both mirin and PD166285, the hyper-phosphorylated forms of APC3 and MCM4 were maintained even at 20 min, and the phosphorylation of CDK1 at Thr14 and Tyr15 did not increase, indicating that the effect of mirin on CDK inactivation is mediated by Wee1/Myt1 kinase activities (lanes 7–9). By contrast, APC3 and MCM4 became hypo-phosphorylated at 10 min in the presence of both p27 and PD166285, although the phosphorylation of CDK1 at Thr14 and Tyr15 was suppressed (lanes 13–15), showing that p27 can directly

inactivate CDK activity irrespective of its phosphorylation status. When S-extract or M-extract was incubated without nuclei, the addition of mirin did not show any inhibitory effect on APC3, MCM4, or CDK1 (Fig 3B, lanes 16–19), eliminating the possibility that mirin directly inhibits CDK. Furthermore, immunoblotting of the chromatin fractions showed that the addition of PD166285 nullified the inhibitory effect of mirin on mitotic entry and MRD (Fig 3C, lanes 3–6). These results suggest that when stalled replication forks are left without Mre11-dependent processing during the early stages of mitotic entry, Wee1/Myt1 activities remain high to inhibit the establishment of a mitotic state by inactivating mitotic CDK activity, causing the cell cycle to return to interphase.

We also examined the ATR-Chk1 checkpoint activity by immunoblotting of nuclear fractions (Fig 3D and E). During interphase, the addition of aphidicolin led to the mobility shift and phosphorylation of Chk1, indicating the activation of the ATR-Chk1 pathway (lanes I and II). After the addition of M-extract, the active form of Chk1 almost disappeared at 40 min (lanes III). Without mitotic induction, the addition of mirin in the second incubation significantly suppressed further accumulation of the active form of Chk1 at 20 min (lanes IV, $P$ = 0.0426), suggesting the role of Mre11 in checkpoint activation during interphase as previously described (Lee & Dunphy, 2013). However, the remaining active form of Chk1 was maintained even after mitotic induction (lanes V). This suggests that Mre11 exonuclease activity is necessary for attenuation of the ATR-Chk1 pathway upon mitotic entry. As Chk1 suppresses Cdc25-mediated removal of the inhibitory phosphorylation of CDK1 Thr14 and Tyr15, it is likely that both ATR-Chk1 and Wee1/Myt1 pathways cooperate in the inactivation of mitotic CDK in the absence of Mre11 exonuclease activity.

## A nonphosphorylatable CDK1 mutant can promote mitotic entry without Mre11 exonuclease activity

To further clarify the role of ATR-Chk1 and Wee1/Myt1 pathways in inactivating mitotic CDK in the absence of Mre11 exonuclease activity, we used wild type and a nonphosphorylatable mutant of recombinant GST-CDK1. In the mutant, Thr14 and Tyr15 were substituted with Ala and Phe, respectively (designated as CDK1-AF in comparison with wild–type CDK1-WT). These CDK1 proteins were preincubated with recombinant protein MBP–cyclin B1-ΔN90 (a nondegradable mutant from the deletion of the N-terminal 90-amino acid region that contains the destruction box) to form active CDK1–cyclin B1 complexes.

The time course of mitotic entry and progression was monitored by the immunoblotting of the whole extracts (Fig 4A and B). When DNA replication was almost complete without aphidicolin (Fig 4A, –Aph 80 min), both CDK1-WT and CDK1-AF were able to promote mitotic entry as indicated by the appearance of hyper-phosphorylated forms of APC3 and MCM4 at 40 or 60 min and the disappearance of Thr14 and Tyr15 phosphorylation of endogenous CDK1 at 40 and 60 min (Fig 4A, lanes 1–8). GST-CDK1-WT was slightly and transiently phosphorylated at 20 min but was dephosphorylated together with endogenous CDK1 at later time points. By contrast, in the presence of aphidicolin, CDK1-WT could not promote mitotic entry, and both recombinant and endogenous CDK1 were more strongly phosphorylated at later time points, whereas CDK1-AF was able to promote mitotic entry in a similar

fashion to the case without aphidicolin (Fig 4A, lanes 9–16). These results show that inhibitory phosphorylation of CDK1 Thr14 and Tyr15 is predominant in the presence of stalled replication forks and that the nonphosphorylatable mutant CDK1 can overcome this inhibitory effect.

We then tried to induce mitotic entry using CDK1-WT/AF in the presence of mirin and aphidicolin (Fig 4B). CDK1-WT was unable to induce mitotic entry in the presence or absence of mirin (Fig 4B left, lanes 1–4, lanes 9–12). CDK1-AF seemed to partly promote mitotic entry even in the presence of mirin as indicated by the observation that APC3 became gradually phosphorylated and nearer to the hyper-phosphorylated state during the experiment, whereas most MCM4 belatedly turned into the hyper-phosphorylated form at 60 min (Fig 4B left, lanes 5–8, lanes 13–16). The phosphorylation of endogenous CDK1 remained at 40 min but disappeared at later time points, providing additional evidence of mitotic entry. Immunoblotting of the chromatin fractions also showed that CDK1-AF, but not CDK1-WT, was able to promote mitotic entry and MRD even in the presence of mirin, although both processes were less efficient than in the absence of mirin (Fig 4C, lanes 1–8). These results suggest that the inhibitory phosphorylation of CDK1 Thr14 and Tyr15 is largely responsible for the inability of CDK1-WT to promote mitotic entry without Mre11 exonuclease activity and that mirin has additional inhibitory effects on mitotic entry and progression other than the phosphorylation of CDK1 Thr14 and Tyr15.

Finally, we examined the effect of PD166285 treatment on recombinant CDK1–cyclin B1–driven mitotic entry in the presence of aphidicolin with or without mirin. Because mitotic induction by the recombinant proteins took longer than M-extract, we added PD166285 at the beginning of the first incubation. Immunoblotting of whole extracts (Fig 4B) and chromatin fractions (Fig 4C) showed that the PD166285 treatment accelerated the timing of APC3 hyper-phosphorylation by CDK1-AF and enabled CDK1-WT to induce mitotic entry and MRD at a comparable level with CDK1-AF in the absence of mirin (Fig 4B right, lanes 5–10 versus left, lanes 5–8; Fig 4C, lanes 9–12 versus lanes 1–4). Thus, inhibition of Wee1/Myt1 was sufficient to promote mitotic entry and MRD when Mre11 exonuclease was functional. Importantly, PD166285 enabled CDK1-WT to promote mitotic entry and MRD even in the presence of mirin (Fig 4B right, lanes 11–13 versus left, lanes 9–12; Fig 4C, lanes 13–14 versus lanes 5–6). The timing of APC3 hyper-phosphorylation by CDK1-AF was also accelerated by PD166285 (Fig 4B right, lanes 14–16 versus left, lanes 13–16). This suggests that Wee1/Myt1–dependent inhibitory phosphorylation of CDK1 Thr14 and Tyr15 plays a significant role in suppressing mitotic entry and MRD when Mre11 exonuclease is not functional.

However, it should be noted that both mitotic entry and MRD were less efficient in the presence of mirin than in the absence of mirin (Fig 4B right, lanes 11–16 versus lanes 5–10; Fig 4C, lanes 13–16 versus lanes 9–12). Specifically, APC3 was not fully phosphorylated in the presence of mirin even with CDK1-AF or PD166285. In addition, CDK1-AF was less active than CDK1-WT in the presence of both PD166285 and mirin (Fig 4B right lanes 11–13 versus lanes 14–16; Fig 4C, lanes 13–14 versus lanes 15–16). It is possible that unprocessed stalled forks without Mre11 exonuclease continue activating ATR-Chk1 or other pathways to suppress mitotic entry independently of the regulation of CDK1 phosphorylation. Alternatively, Mre11 exonuclease may be involved in another process that is required for

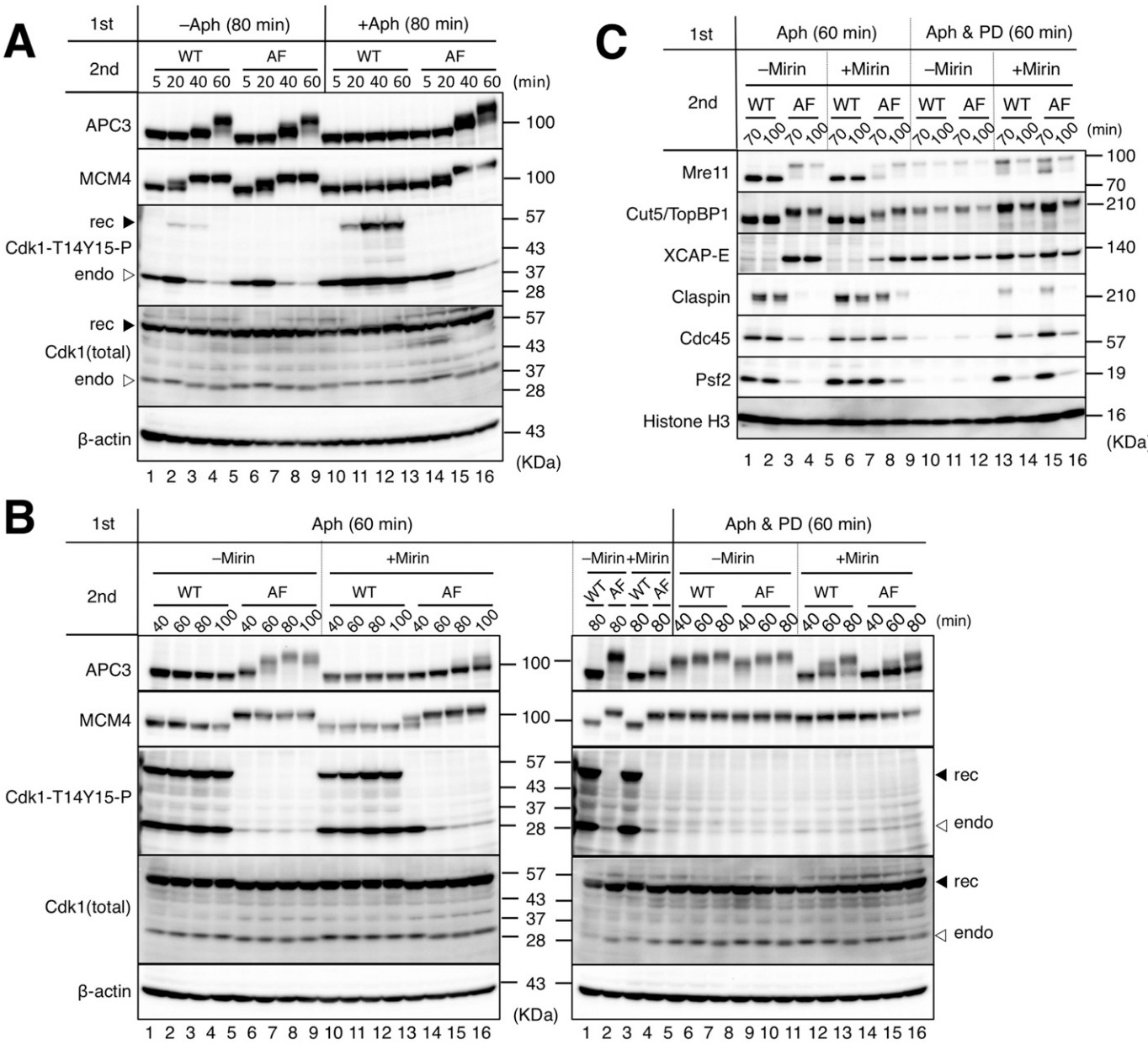

**Figure 4. A nonphosphorylatable CDK1 mutant can promote mitotic entry without Mre11 exonuclease activity.**
**(A)** The ability of the recombinant CDK complexes of MBP–cyclin B1-ΔN90 with GST-CDK1 wild type (CDK-WT, WT) or GST-CDK1-T14A/Y15F (CDK-AF, AF) to promote mitotic entry in the absence or presence of replication stress. In the first reaction, sperm nuclei (5,000/μl) were incubated for 80 min in 20 μl of S-phase egg extract with (+Aph) or without (−Aph) 10 μg/ml of aphidicolin. In the second reaction, CDK-WT or CDK-AF was added at 2.2 μM and incubated for the indicated times (5, 20, 40, and 60 min). 1 μl of the whole extracts were sampled and analyzed by immunoblotting. The positions of recombinant GST-CDK1 (rec) and endogenous CDK1 (endo) are indicated by filled and open triangles, respectively. **(B, C)** Restoration of mitotic entry by the nonphosphorylatable CDK1 mutant. **(A)** A similar experiment with that shown in (A) was performed with the indicated conditions. PD166285 (PD) and mirin were added at 5 and 100 μM, respectively. **(B, C)** The whole extract (B) and the chromatin fractions (C) were analyzed by immunoblotting.
Source data are available for this figure.

full activation of the APC/C complex. As the H1 kinase activity of the CDK1-Y15F mutant was lower than that of CDK1-WT in the absence of Wee1 phosphorylation (Mueller et al, 1995), the difference in promotion of mitotic entry between CDK1-WT and AF may reflect their intrinsic kinase activity. Overall, these results suggest that Mre11 exonuclease have multiple roles in promoting mitotic entry in the presence of stalled forks.

## Discussion

In this study, we discovered that Mre11 exonuclease activity promotes mitotic entry in the presence of stalled forks. This activity is not required for mitotic entry when DNA replication is complete. Mre11 exerts its function during an initial short period of mitotic entry, and in the absence of Mre11 activity, mitotic CDK activity is

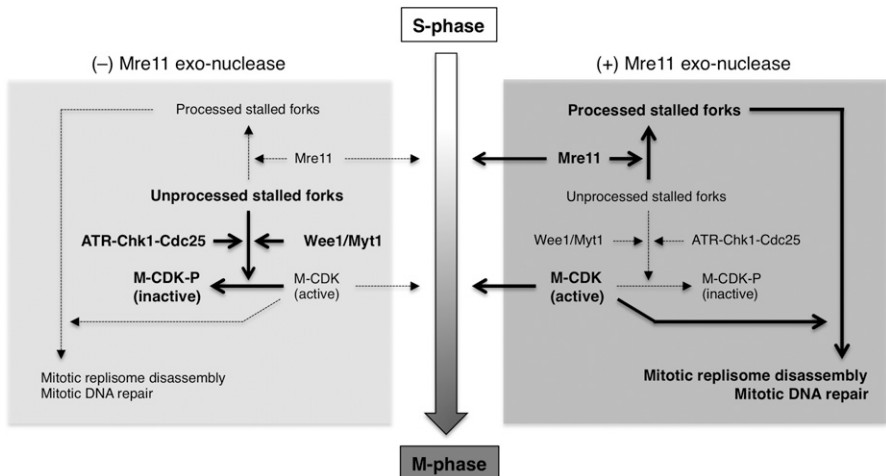

**Figure 5.  A model for mitotic entry in the presence of stalled forks.**
Active pathways are indicated with bold font/lines and large arrows, whereas inactive pathways are indicated with regular font/dotted lines and small arrows.

inactivated by Wee1/Myt1–dependent inhibitory phosphorylation, with the cell cycle returning to interphase. A nonphosphorylatable CDK mutant was able to partially bypass the requirement of Mre11 for mitotic entry. From these results, we propose a model in which Mre11 facilitates the processing of stalled replication forks upon mitotic entry, attenuating the inhibition of mitotic CDK activation and promoting mitotic progression and concomitant mitotic events, such as MRD and mitotic DNA repair synthesis (MiDAS) (Fig 5).

We showed that the action of Mre11 after mitotic induction was necessary and sufficient for subsequent mitotic progression and MRD (Fig 2), raising questions as to why Mre11 cannot work during S-phase and what triggers the action of Mre11 upon mitotic induction. We previously reported that Rad51 protects nascent DNA from unusual Mre11-dependent degradation during the S-phase (Hashimoto et al, 2010), and this function of Rad51 is now widely accepted as an important part of the fork protection mechanism involving many types of DNA repair factors, such as BRCA1/2, FANCD2, Mus81, and CtIP (Schlacher et al, 2011; Kolinjivadi et al, 2017; Taglialatela et al, 2017). Therefore, if the role of Mre11 in promoting mitotic entry were to degrade nascent DNA at stalled forks, it would be reasonable that this Mre11 function is suppressed by the fork protection mechanism during the S-phase. In this case, mitotic induction may immediately attenuate the fork protection mechanism, possibly through increased CDK activity within nuclei before nuclear envelope breakdown, allowing Mre11 to act on nascent DNA and cause irreversible mitotic entry. In support of this, it is known that the C-terminal Rad51 binding domain of BRCA2 becomes phosphorylated in a CDK-dependent manner during G2/M transition, resulting in disassembly of the Rad51 filament (Esashi et al, 2005; Ayoub et al, 2009). Another possibility is that Mre11 activity itself is regulated by mitotic CDK. In fact, Mre11 is highly phosphorylated in the M-phase (Figs 1–4). In future studies, it will be important to clarify why and how Mre11 function is suppressed in the interphase and activated upon mitotic entry.

We then sought to determine what type of fork structure is targeted by Mre11. We showed that Mre11 exonuclease activity, but not endonuclease activity, is necessary for mitotic entry only when DNA replication progression is inhibited (Figs 1 and 2). In the case of DSB end processing for homologous recombination, Mre11

endonuclease activity initially nicks the 5′-terminated strand at an internal site, and Mre11 exonuclease activity then resects the nicked strand in the 3′-5′ direction to produce a 3′-overhang, which is coated with the Rad51 filament (Paull, 2018; Syed & Tainer, 2018; Reginato & Cejka, 2020). This process is completely suppressed by the endonuclease inhibitor PFM01 as the initial nicking activity is essential (Moiani et al, 2018). By contrast, the exonuclease activity, independently from the endonuclease activity, is responsible for Mre11–dependent fork degradation that occurs in the absence of a fork protection mechanism (e.g., in BRCA1/2–defective cells) (Schlacher et al, 2011; Kolinjivadi et al, 2017; Taglialatela et al, 2017). The entry point of Mre11 for uncontrolled fork degradation is assumed to be the regressed arms of reversed replication forks (Kolinjivadi et al, 2017; Taglialatela et al, 2017; Rickman & Smogorzewska, 2019). Although it is unclear what proportion of stalled forks were actually converted into reversed forks in our experiments, it is possible that the role of Mre11 in promoting mitotic entry is to process reversed forks. In this regard, it will be interesting to examine the involvement of fork remodeling enzymes such as SMARCAL1, HLTF, and ZRANB3 in mitotic entry under replication stress (Poole & Cortez, 2017).

Mitotic CDK activity was rapidly inactivated by Wee1/Myt1–dependent phosphorylation in the presence of mirin and replication stress (Fig 3), suggesting that Wee1/Myt1 kinase activities are maintained at a high level in the persistent presence of unprocessed, stalled (or maybe reversed) forks after mitotic induction. Although it is unclear whether Wee1/Myt1 is actively regulated under these conditions, the ATR-Chk1–dependent checkpoint pathway should also have a significant role in the suppression of mitotic CDK activity (Saldivar et al, 2017). It was reported that Mre11 activity is required for full activation of Chk1 under replication stress during interphase (Lee & Dunphy, 2013), and we also observed that Chk1 activation did not proceed efficiently after the addition of mirin without mitotic induction (Fig 3), suggesting a positive role of Mre11 in promoting checkpoint activation. However, the activated form of Chk1 was maintained in the presence of mirin during the interphase and even after mitotic induction (Fig 3), which likely cooperates with Wee1/Myt1 to inactivate mitotic CDK activity. Therefore, Mre11 may also function in checkpoint attenuation or

recovery by degrading the fork structure necessary for ATR activation during an initial period of mitotic entry. In light of the findings that inhibition of Wee1/Myt1 and recombinant CDK1-AF can bypass the requirement of Mre11 for both mitotic entry and MRD (Figs 3 and 4), it is likely that the primary role of Mre11 exonuclease activity is to regulate mitotic CDK activity and that this exonuclease activity itself is not required for MRD, although there is still the possibility that the exonuclease activity of Mre11 is necessary for subsequent MiDAS. We previously detected MiDAS-like activity in the experiment in which replication forks stalled by ara-CTP were released by excess dCTP after mitotic entry (Hashimoto & Tanaka, 2021). Although we performed this experiment repeatedly, the addition of excess dCTP often caused mitotic exit in the presence of mirin but not in the absence of mirin (data not shown). This mitotic exit was not suppressed by PD166285 or CDK1-AF (data not shown). Therefore, it is currently unclear whether Mre11 exonuclease promotes MiDAS.

We observed that both CDK1-WT/PD166285 and CDK1-AF were not able to fully phosphorylate APC3 in the presence of mirin (Fig 4). This raises several possibilities. First, unprocessed stalled forks (or reversed forks) may affect other regulatory processes for CDK1 beyond inhibitory phosphorylation. Second, Mre11 has specific mitotic roles other than DNA processing, which may affect the observed inefficient phosphorylation of APC3 in the presence of mirin. It was reported that the MRN complex regulates spindle assembly (Rozier et al, 2013) and spindle dynamics in mitosis (Xu et al, 2018). Specifically, the former function is dependent on Mre11 exonuclease activity, which is required for RCC1 chromatin association and the subsequent establishment of a RanGTP gradient around chromatin (Rozier et al, 2013). Therefore, the spindle assembly checkpoint pathway may be activated to suppress the full activation of APC3 in the presence of mirin.

Finally, we utilized a unique system based on *Xenopus* egg extracts to reveal a novel role of Mre11 for mitotic entry under replication stress. In the future, it will be important to examine whether this function of Mre11 is conserved among other species, especially in mammals. It is possible that Mre11-dependent fork processing may frequently occur upon mitotic entry during unperturbed cell cycles at difficult-to-replicate regions such as common fragile sites and centromeres, and its failure may promote genomic instability.

# Material and Methods

### *Xenopus laevis* egg extracts, chromatin fractions, nuclear fractions, and chemicals

S-phase (interphase) and M-phase (CSF-arrested) egg extracts and de-membranated sperm nuclei were prepared as previously described (Murray & Kirschner, 1989; Kubota & Takisawa, 1993). Two or three different preparations were mixed to normalize the activities of egg extracts. In all of the experiments using egg extract, the reaction temperature was 23°C, and the concentration of sperm nuclei was 5,000 nuclei/µl in the first reaction with the S-phase extract. This was diluted to 2,500 nuclei/µl in the second reaction when mitotic entry was induced with an equal volume of the M-phase extract. In the experiment of Fig 2C as an exception, the first reaction was diluted with 1.5 volumes of S-phase extract and

2.5 volumes of M-phase extract to produce 1,000 nuclei/µl in the second reaction.

To isolate chromatin fractions, 20 µl (S-phase) or 40 µl (S-phase and M-phase) of egg extract (containing a total of 100,000 nuclei) was diluted with 700 µl of EB buffer (100 mM KCl, 2.5 mM MgCl₂, and 50 mM Hepes–KOH, pH 7.5) containing 0.2% NP40 and layered onto 200 µl of a 30% (wt/vol) sucrose cushion made with the same buffer. The chromatin was centrifuged at 13,200$g$ for 5 min at 4°C, washed with 300 µl of EB buffer, and centrifuged again at 20,000$g$ for 1 min. In the experiment of Fig 2C as an exception, 100 µl of egg extract was initially diluted with 1 ml of EB + 0.2% NP40, and the chromatin was washed with 500 µl of EB. The pellet was resuspended in SDS–PAGE sample buffer and analyzed by immunoblotting. To examine the phosphorylation of chromatin–bound proteins (Fig 1C), the chromatin pellet was resuspended in 40 µl of alkaline phosphatase buffer (1 mM MgCl₂, 1 mM PMSF, and 50 mM Tris–HCl, pH 9.0) with or without six units of calf intestine alkaline phosphatase (TOYOBO), incubated for 30 min at 30°C, washed with 500 µl of EB buffer, centrifuged again at 20,000$g$ for 1 min, and analyzed by immunoblotting.

To isolate nuclear fractions, egg extract with 100,000 nuclei was diluted with 500 µl of EB buffer and layered onto 300 µl of EB plus 30% (wt/vol) sucrose buffer. The nuclei were centrifuged at 6,620$g$ for 2 min at 4°C, washed with 1 ml of EB plus 30% (wt/vol) sucrose buffer, and centrifuged again at 6,620$g$ for 2 min. The pellet was resuspended in SDS–PAGE sample buffer and analyzed by immunoblotting.

The following chemicals were used at the indicated concentrations: aphidicolin (10 µg/ml; Sigma-Aldrich), mirin (20, 50 or 100 µM; Sigma-Aldrich), PFM01 (50 or 100 µM; Sigma-Aldrich), NMS-873 (100 µM; Sigma-Aldrich), MLN-4924 (10 µM; Sigma-Aldrich), and PD166285 (5 or 10 µM; Sigma-Aldrich).

### cDNA cloning and primers

cDNA encoding wild-type full-length *X. laevis* CDK1B (CDK1-WT) was first amplified by PCR with the primers 5′-BamHI-xCDK1B and xCDK1B-HindIII-3′ using *X. laevis* oocyte cDNA as a template. cDNA encoding the CDK1-AF mutant (in which Thr14 and Tyr15 of CDK1B were replaced with Ala and Phe, respectively) was then amplified by PCR with primers 5′-BamHI-xCDK1B-AF and xCDK1B-HindIII-3′ using the PCR product of CDK1-WT as a template. cDNA encoding *Saccharomyces cerevisiae* Civ1, a CDK-activating kinase, was also amplified by PCR with primers 5′-HindIII-EcoRI-Civ1 and Civ1-EagI-3′ using a plasmid containing the sequence of Civ1 (provided by Dr TS Takahashi, Kyushu University) as a template. The PCR products from the reactions for CDK1-WT and CDK1-AF were digested with BamHI and HindIII, whereas the PCR product from the reaction for Civ1 was digested with HindIII and EagI. The CDK1-WT and CDK1-AF fragments were ligated with the Civ1 fragment in tandem and cloned into pGEX 6P-1 (Cytiva) between the BamHI and EagI (NotI) sites (CDK1-WT-Civ1/pGEX 6P-1 and CDK1-AF-Civ1/pGEX 6P-1) for co-expression of Civ1 with GST-tagged CDK1-WT and CDK1-AF in *Escherichia coli*.

cDNA-encoding *X. laevis* cyclin B1 lacking the N-terminal 90 amino acids was amplified by PCR with the primers 5′-BamHI-xcyclin B1-E91 and xcyclin B1-EcoRI-3′ using *X. laevis* oocyte cDNA as a template and cloned into pGEX 6P-1 between the BamHI

and EcoRI sites (cyclin B1-ΔN90/pGEX 6P-1). The fragment containing the cyclin B1-ΔN90 sequence was recovered after BamHI and SalI digestion and cloned into pMAL-C4X (New England Biolabs) between the BamHI and SalI sites (cyclin B1-ΔN90/pMAL-C4X) for expression of MBP-tagged cyclin B1-ΔN90 in *E. coli*.

cDNA encoding full-length *X. laevis* Mre11 was amplified by PCR with primers 5′-BamHI-xMre11 and xMre11-EcoRI-3′ using *X. laevis* oocyte cDNA as a template and cloned into pHEX-1 (a modified pGEX 6P-1 vector in which the GST–encoding sequence was replaced with the His8-encoding sequence: provided by Dr S Mochida, Kumamoto University) between the BamHI and EcoRI sites for expression of His–tagged Mre11 in *E. coli*.

The sequences of the PCR primers are as follows (single and double underlines show the restriction sites and mutation site of CDK1-AF, respectively):

5′-BamHI-xCDK1B (5′-TAT<u>GGATCC</u>GATGGACGAGTACACTAAAATAGAG-3′)
5′-BamHI-xCDK1B-AF (5′-TAT<u>GGATCC</u>GATGGACGAGTACACTAAAATA-GAGAAGATTGGAGAGGGC<u>G</u>CATTTGGGGTCGTG-3′)
xCDK1-HindIII-3′ (5′-TAT<u>AAGCTT</u>GTTTTAATTTCTAATCTGATTGGCG-3′)
5′-HindIII-EcoRI-Civ1 (5′-TAT<u>AAGCTTGAATTC</u>ATGAAACTGGATAGTATAG-3′)
Civ1-EagI-3′ (5′-TAT<u>CGGCCG</u>TTATGGCTTTTCTAATTCTTGCAAG-3′)
5′-BamHI-xcyclin B1-E91 (TAT<u>GGATCC</u>GAACCCAGCTCACCAAGCCCAATG)
xcyclin B1-EcoRI-3′ (5′-TAT<u>GAATTC</u>TCATACAAGTGGGCGGGCCATTGC-3′)
5′-BamHI-xMre11 (5′-TAT<u>GGATCC</u>ATGAGTTCTTCTAGTAGCTCTCTG-3′)
xMre11-EcoRI-3′ (5′-TAT<u>GAATTC</u>TTATCTACGGCCCCTTCTGGAAGG-3′)

### Protein purification

GST-CDK1-WT or GST-CDK1-AF was co-expressed with untagged Civ1 in Rossetta(DE3)pLysS (Novagen) to induce CDK1 phosphorylation at Thr161, which is required for CDK1 activity. The GST-tagged proteins were then purified with Glutathione Sepharose 4B (Cytiva) according to the manufacturer's protocol. MBP–cyclin B1-ΔN90 was expressed in Rossetta(DE3)pLysS and purified with amylose resin (New England Biolabs) according to the manufacturer's protocol. The purified proteins were concentrated and buffer exchanged to PBS(−) using an Amicon Ultra-4 30K (Merck) ultrafiltration device. The protein concentrations of all samples were then adjusted to 22 $\mu$M. GST-CDK1-WT or GST-CDK1-AF was mixed with an equal volume of MBP–cyclin B1-ΔN90 and incubated for 15 min at 23°C to promote the formation of the CDK1–cyclin B1 complex. These mixtures were added to the egg extract at 2.2 $\mu$M to drive mitotic entry.

His-Mre11 was expressed in BL21-CodonPlus(DE3)-RIPL (Agilent Technologies) and purified with Ni-NTA agarose (QIAGEN) according to the manufacturer's protocol under denaturing conditions using 8 M urea. The purified protein was then dialyzed with PBS(−), and the resultant precipitate was used as an antigen to immunize rabbits. Anti-Mre11 polyclonal antibodies were affinity purified with the antigen immobilized on Affi-Gel15 (Bio-Rad). His-p27 was prepared as previously described (Hashimoto et al, 2010) and used at 100 $\mu$g/ml to inhibit CDK activity.

### Immunoblotting and antibodies

SDS–PAGE and immunoblotting were performed according to standard procedures. The chemiluminescent signals were detected with ChemiDoc XRS+ (Bio-Rad) and quantified with Image J software. The primary antibodies or antiserum were used at the indicated dilutions as follows: rabbit polyclonal anti-Mre11 antibody (174 $\mu$g/ml, 1:200) was prepared in this study; rabbit polyclonal antibodies to Psf2 (183 $\mu$g/ml, 1:200) and XCAP-E (936 $\mu$g/ml, 1:500) were previously described (Hashimoto & Tanaka, 2018); rabbit antisera to Cut5/TopBP1 (1:3,000), Cdc45 (1:2,000), claspin (1:1,000), Pol$\varepsilon$ (p60; 1:1,000), Sld5 (1:300), and Chk1 (1:400) were provided by Y. Kubota (Osaka University) and H. Takisawa (Osaka University); rabbit antiserum to Pol$\delta$ (p66) (1:1,000) was provided by S. Waga (Japan Women's University); rabbit antiserum to Cdk1 (1:500), mouse monoclonal anti–cyclin B2 antibody (X121, 0.5 mg/ml, 1:200), and mouse monoclonal anti–APC3/CDC27 (610455, 1:200; BD Biosciences) were provided by S. Mochida (Kumamoto University). The following antibodies were obtained from the indicated companies: rabbit polyclonal antibodies to SMC1 (4802, 1:1,000; Cell Signaling); phospho-CDK1 (Tyr15; 9111, 1:500; Cell Signaling); phospho-CDK1 (Thr14, Tyr15; 44-686G, 1:500; Thermo Fisher Scientific); phospho-Chk1 (Ser345; 2341, 1:300; Cell Signaling); MCM4 (ab4459, 1:2,000; Abcam); rabbit monoclonal anti-histone H3 antibody (4499, 1:800; Cell Signaling); mouse monoclonal antibodies to MCM7 (sc-9966, 1:2,500; Santa Cruz); and $\beta$-actin (ab8224, 1:5,000; Abcam).

## Supplementary Information

## Acknowledgements

We thank H Takisawa, Y Kubota, S Waga, S Mochida, and TS Takahashi for providing antibodies/antisera (to Cut5/TopBP1, claspin, Cdc45, Chk1, Pol$\varepsilon$ [p60], Pol$\delta$ [p66], APC3, and cyclin B2), pHEX-1 vector, and Civ1 cDNA. This work was supported by JSPS KAKENHI grants to Y Hashimoto (19K06617 and 15K06855) from the Ministry of Education, Cultures, Sports, Science and Technology (MEXT) in Japan.

### Author Contributions

Y Hashimoto: conceptualization, data curation, funding acquisition, investigation, methodology, project administration, and writing—original draft, review, and editing.
H Tanaka: supervision.

### Conflict of Interest Statement

The authors declare that they have no conflict of interest.

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
