## [Reviewer comments · Life Science Alliance]

Life Science Alliance

Mre11 exonuclease activity promotes irreversible mitotic progression under replication stress

Yoshitami Hashimoto and Hirofumi Tanaka

DOI: <https://doi.org/10.26508/lsa.202101249>

Corresponding author(s): Yoshitami Hashimoto, Tokyo University of Pharmacy and Life Sciences

Review Timeline:

Submission Date:	2021-09-29
Editorial Decision:	2021-10-26
Revision Received:	2022-01-14
Editorial Decision:	2022-02-01
Revision Received:	2022-02-26
Editorial Decision:	2022-03-01
Revision Received:	2022-03-02
Accepted:	2022-03-02

Transaction Report:

October 26, 2021

Re: Life Science Alliance manuscript #LSA-2021-01249-T

Dr. Yoshitami Hashimoto
Tokyo University of Pharmacy and Life Sciences
School of Life Sciences
1432-1
Horinouchi
Hachioji, Tokyo 192-0392
Japan

Dear Dr. Hashimoto,

Thank you for submitting your manuscript entitled "Mre11 exonuclease activity promotes irreversible mitotic progression under replication stress" to Life Science Alliance. The manuscript was assessed by expert reviewers, whose comments are appended to this letter. We invite you to submit a revised manuscript addressing the Reviewer comments.

Thank you for this interesting contribution to Life Science Alliance. We are looking forward to receiving your revised manuscript.

Sincerely,

B. MANUSCRIPT ORGANIZATION AND FORMATTING:

Reviewer #1 (Comments to the Authors (Required)):

This paper uses *Xenopus* egg extract to show that Mre11 exonuclease activity is required to promote mitotic entry in the presence of stalled forks. In the absence of Mre11, mitotic entry is blocked by Wee1/Myt1 activity that inhibits CDK1 activity. This is an interesting observation, though there is not a great deal of mechanistic insight into exactly what Mre11 is doing. One complication is that the essential activity of Mre11 for mitotic entry cannot be performed during interphase and so appears to require high CDK activity, creating a positive feedback loop. This is a relatively simple set of observations and it is easy to think of additional experiments to unravel the mechanism a bit more - for example, is the ATR-Chk1-Cdc25 pathway essential for the inhibition of mitotic entry, and is Mre11 phosphorylation by CDKs essential for its pro-mitotic activity. However, I think the data and conclusions drawn from it are fairly clear and are adequately described in the text and no additional experiments are required.

One relatively small problem is where the text in the last paragraph of page 13 says that '...the PD166285 treatment caused CDK1-WT to induce mitotic entry and MRD at comparable level with CDK1-AF both in the absence and presence of mirin..' That doesn't seem quite right: in Fig 4B lanes 11-16, in the presence of PD and Mirin, the AF mutant is less active than WT CDK1 in promoting APC3 phosphorylation; a similar effect is seen in Fig 4C, lanes 13-16 where the AF mutant is less active than WT CDK1 in promoting Mre11 phosphorylation. The text should be amended to reflect the data in the figure. What might cause this? Later on in the paragraph the authors speculate that Mirin might decrease CDK activity by other pathways in addition to T14/Y15 phosphorylation, but why would this preferentially affect the AF mutant?

Minor Point

Ref 30 is cited for use of araCTP in egg extract, but it seems to be wrong (a review of TRAIP) - probably should be Walter and Newport 1997?

Reviewer #2 (Comments to the Authors (Required)):

In this manuscript, the authors describe a novel role for Mre11 exonuclease activity in promoting mitotic entry in the presence of stalled forks. In particular, they show that Mre11 exonuclease activity facilitates the processing of stalled replication forks attenuating the inhibition of mitotic CDK activation. This in turn allows mitotic progression and replisome disassembly in the presence of prolonged fork stalling.

This novel pathway provides an interesting physiological explanation for the poorly understood Mre11 mediated nascent DNA processing that the lead author discovered a few years ago.

The experiments performed are clear and well-controlled, supporting the major claims. I am surprised that the activity of the ATR-CHK1 checkpoint in the presence and absence of Mre11 inhibitors has not been measured. This is relatively easy to do using commercially available anti-phospho Chk1 antibodies that are widely used by other authors in *Xenopus* egg extracts. I think this is an issue that could be addressed to further consolidate this nice paper. Also, I suggest clarifying the abstract to make it more accessible to non-specialists.

Reviewer #3 (Comments to the Authors (Required)):

In this study, Hashimoto and Tanaka use *Xenopus* egg extracts to study mitotic entry and replisome disassembly at stalled replication forks. By using protein inhibitors, they conclude that the Mre11 nuclease is required to process stalled replication forks in order activate mitotic CDK and promote mitotic entry.

Although the presented data are of excellent quality, they bring very few conceptual advances compared to the previous

publications of the authors. Throughout the manuscript, the results are only based on the addition of an M-phase extract on top of a S-phase extract to force mitotic entry, and the intrinsic design of the experiments thus limits the strength of the conclusions that could be made and the generalization to other models.

Nevertheless, the discovery that Mre11 appears to be necessary for mitotic entry after fork stalling is very interesting and could be substantiated by additional experiments.

1. In the absence of Aphidicolin, the replisome components are almost completely unloaded from chromatin at the end of the 2nd incubation (Fig 1D, lane 1). Does the addition of mirin alone (without M-phase extract) in the 2nd incubation impede this unloading? It would involve Mre11 in the processing of other structures than stalled forks.
2. In the same line, what would happen if mirin was added during the first incubation with aphidicolin? It is expected that the BRCA2-RAD51 pathway should already protect against stalled fork processing by Mre11. Then, would the addition of the M-phase extract still promote mitotic entry and replisome disassembly?
3. Mre11-driven processing of stalled forks may attenuate the checkpoint signaling by ATR and CHK1. It would be a valuable information to show that Mre11 inhibitor does not allow checkpoint attenuation and consequently, the loss of inhibition of mitotic CDK.
4. The authors propose that Mre11 processing of stalled forks at the beginning of mitosis may promote MiDAS. Could the authors test this hypothesis by using the experimental setup they published in Hashimoto and Tanaka, JCB 2020 (Fig 5D; addition of dCTP 20 or 30 minutes after M-phase extract addition)?

Minor remarks:

Page 8: "Mitotic entry was confirmed by the mobility shifts of Mre11 and Cut5/TopBP1". Could the authors explain what are these mobility shifts? And indicate a reference if already described?

Page 9: related to Figure 2A. Please indicate in the figure the size of the DNA fragments containing Cy5-dUTP.

Page 13: related to Figure 4C "CDK1-AF was able to promote mitotic entry and MRD even in the presence of mirin". Please modify this conclusion to indicate clearly that both processes occur less efficiently than in the absence of mirin.

'Referee Cross-Comments'

Taking into account the comments of the other referees, I have decided to remove my first claim that could be technically challenging.

"1. It would be great to show that Mre11 actually processes stalled forks after the addition of the M-phase extract. Could this be revealed by primer extension by T4 DNA polymerase (with a radioactive or fluorescent dNTP) and denaturing gels as performed in Hashimoto et al., NSMB 2010? Or with another assay (DNA fibers)?"

Responses to the reviewers' comments

Reviewer #1 (Comments to the Authors (Required)):

–This paper uses *Xenopus* egg extract to show that Mre11 exonuclease activity is required to promote mitotic entry in the presence of stalled forks. In the absence of Mre11, mitotic entry is blocked by Wee1/Myt1 activity that inhibits CDK1 activity. This is an interesting observation, though there is not a great deal of mechanistic insight into exactly what Mre11 is doing. One complication is that the essential activity of Mre11 for mitotic entry cannot be performed during interphase and so appears to require high CDK activity, creating a positive feedback loop. This is a relatively simple set of observations and it is easy to think of additional experiments to unravel the mechanism a bit more - for example, is the ATR-Chk1-Cdc25 pathway essential for the inhibition of mitotic entry, and is Mre11 phosphorylation by CDKs essential for its pro-mitotic activity. However, I think the data and conclusions drawn from it are fairly clear and are adequately described in the text and no additional experiments are required.

–One relatively small problem is where the text in the last paragraph of page 13 says that '...the PD166285 treatment caused CDK1-WT to induce mitotic entry and MRD at comparable level with CDK1-AF both in the absence and presence of mirin.' That doesn't seem quite right: in Fig 4B lanes 11-16, in the presence of PD and Mirin, the AF mutant is less active than WT CDK1 in promoting APC3 phosphorylation; a similar effect is seen in Fig 4C, lanes 13-16 where the AF mutant is less active than WT CDK1 in promoting Mre11 phosphorylation. The text should be amended to reflect the data in the figure. What might cause this? Later on in the paragraph the authors speculate that Mirin might decrease CDK activity by other pathways in addition to T14/Y15 phosphorylation, but why would this preferentially affect the AF mutant?

We agree with this reviewer. We amended the text and clearly described that CDK1-AF was less active than CDK1-WT in the presence of PD and mirin together with additional explanations (both in results and discussion).

–Minor Point

–Ref 30 is cited for use of araCTP in egg extract, but it seems to be wrong (a review of TRAIP) - probably should be Walter and Newport 1997?

In the revised version, we retracted the original Fig. 2A and 2B because we recognized that replication restart by dCTP in the presence of Mirin sometimes caused mitotic exit. This may be due to the combined effects of mirin and ongoing forks. Accordingly, we deleted the corresponding reference and reorganized all references by author and year of publication instead of a number in square brackets.

Although we replaced the original Fig. 2A, 2B (ara-CTP stalling & dCTP restart) and 2C with the new Fig. 2A (the requirement of Mre11 for interphase replisome disassembly), Fig. 2C (the effects when mirin was present throughout the 1st and 2nd incubation) and Fig. 2B (the same with the original 2C), the main conclusion of this study was not affected.

Reviewer #2 (Comments to the Authors (Required)):

–In this manuscript, the authors describe a novel role for Mre11 exonuclease activity in promoting mitotic entry in the presence of stalled forks. In particular, they show that Mre11 exonuclease activity facilitates the processing of stalled replication forks attenuating the inhibition of mitotic CDK activation. This in turn allows mitotic progression and replisome disassembly in the presence of prolonged fork stalling. This novel pathway provides an interesting physiological explanation for the poorly understood Mre11 mediated nascent DNA processing that the lead author discovered a few years ago.

–The experiments performed are clear and well-controlled, supporting the major claims. I am surprised that the activity of the ATR-CHK1 checkpoint in the presence and absence of Mre11 inhibitors has not been measured. This is relatively easy to do using commercially available anti-phospho Chk1 antibodies that are widely used by other authors in *Xenopus* egg extracts. I think this is an issue that could be addressed to further consolidate this nice paper. Also, I suggest clarifying the abstract to make it more accessible to non-specialists.

As suggested by this reviewer, we examined the activation of the ATR-Chk1 checkpoint

in the presence and absence of mirin (new Fig. 3D). As expected, Chk1 phosphorylation was maintained in the presence of mirin after mitotic induction, suggesting that ATR-Chk1 and Wee1/Myt1 cooperatively suppress mitotic progression when Mre11 exonuclease is not functional.

We also clarified the abstract by deleting unnecessary technical terms and explaining the results more plainly.

Reviewer #3 (Comments to the Authors (Required)):

–In this study, Hashimoto and Tanaka use *Xenopus* egg extracts to study mitotic entry and replisome disassembly at stalled replication forks. By using protein inhibitors, they conclude that the Mre11 nuclease is required to process stalled replication forks in order activate mitotic CDK and promote mitotic entry.

Although the presented data are of excellent quality, they bring very few conceptual advances compared to the previous publications of the authors. Throughout the manuscript, the results are only based on the addition of an M-phase extract on top of a S-phase extract to force mitotic entry, and the intrinsic design of the experiments thus limits the strength of the conclusions that could be made and the generalization to other models.

–Nevertheless, the discovery that Mre11 appears to be necessary for mitotic entry after fork stalling is very interesting and could be substantiated by additional experiments.

–1. In the absence of Aphidicolin, the replisome components are almost completely unloaded from chromatin at the end of the 2nd incubation (Fig 1D, lane 1). Does the addition of mirin alone (without M-phase extract) in the 2nd incubation impede this unloading? It would involve Mre11 in the processing of other structures than stalled forks.

The addition of mirin alone (without M-phase extract) in the 2nd means to address whether mirin impedes replisome disassembly during normal replication termination in S-phase. In the absence of aphidicolin, replication termination will occur everywhere during the 1st incubation before adding mirin. Therefore, we examined replisome

disassembly in the S-phase extract without aphidicolin when mirin was present throughout the experiment, and found that Mre11 exonuclease activity was not required for replisome disassembly during termination in S-phase (new Fig. 2A).

–2. In the same line, what would happen if mirin was added during the first incubation with aphidicolin? It is expected that the BRCA2-RAD51 pathway should already protect against stalled fork processing by Mre11. Then, would the addition of the M-phase extract still promote mitotic entry and replisome disassembly?

As suggested, we examined the effects of mirin when added from the first incubation, and found that mitotic entry and replisome disassembly were suppressed (new Fig. 2C). This is reasonable because Mre11 has no opportunity to work throughout the 1st and 2nd incubations. However, this question also raises another question as to whether Mre11 can execute its function during the 1st incubation in the absence of the BRCA2-RAD51 pathway. We will examine this issue in a future study.

–3. Mre11-driven processing of stalled forks may attenuate the checkpoint signaling by ATR and CHK1. It would be a valuable information to show that Mre11 inhibitor does not allow checkpoint attenuation and consequently, the loss of inhibition of mitotic CDK.

As suggested, we examined the ATR-Chk1 checkpoint activity. As expected, Chk1 phosphorylation was maintained in the presence of the Mre11 inhibitor even after adding M-extract (new Fig. 3D), suggesting that Mre11 promotes checkpoint attenuation.

–4. The authors propose that Mre11 processing of stalled forks at the beginning of mitosis may promote MiDAS. Could the authors test this hypothesis by using the experimental setup they published in Hashimoto and Tanaka, JCB 2020 (Fig 5D; addition of dCTP 20 or 30 minutes after M-phase extract addition)?

We performed this experiment repeatedly to test the hypothesis, and realized that the restart of DNA synthesis by dCTP often caused mitotic exit in the presence of mirin whenever (at 0 min, 20 min or 30 min) dCTP was added after M-extract addition. Therefore, we decided to retract the original Fig. 2A and 2B. The original Fig. 2A showed

the replication activity in the presence of Mirin and dCTP (added at 0 min), and the data were reproducible, but this was expected because similar activities were observed both in S-phase (+p27) and in M-phase (–p27). In the original Fig. 2B, XCAP-E associated with chromatin later in the presence of both mirin and dCTP (added at 0 min), demonstrating that the cell cycle proceeded to M-phase later than usual. However, this did not always occur and the cell cycle often returned (or proceeded) to interphase. The combination of mirin and dCTP may cause mitotic exit due to the multiple roles of Mre11 in mitosis and ongoing replication forks (that delay mitotic entry, Hashimoto and Tanaka, JBC 2021).

As we previously reported (Hashimoto and Tanaka, JBC 2021), MiDAS-like activity was observed when dCTP was added 20 or 30 min after M-extract addition in the absence of mirin, during which the mitotic state (APC3 hyper-phosphorylation) was maintained. To examine whether this activity was dependent on Mre11, we designed several experiments in which mitotic entry was not suppressed in the presence of mirin using PD166285 or CDK1-AF. This is described in Fig.3 and Fig.4, but the addition of dCTP (at 20 or 30 min) caused mitotic exit even after mitotic entry. Although we were able to detect DNA synthesis activity in this condition, it is unclear whether this activity is MiDAS-like because of mitotic exit. Restart of DNA synthesis by dCTP induced mitotic exit in the presence of mirin, but not the exit in the absence of mirin. We would like to set this issue aside because of time limitations.

–Minor remarks:

–Page 8: "Mitotic entry was confirmed by the mobility shifts of Mre11 and Cut5/TopBP1". Could the authors explain what are these mobility shifts? And indicate a reference if already described?

We examined whether these mobility shifts were dependent on phosphorylation by phosphatase treatment, and added the new Fig. 1C showing that these shifts are dependent on phosphorylation exclusively (Mre11) or partly (Cut5/TopBP1).

–Page 9: related to Figure 2A. Please indicate in the figure the size of the DNA fragments containing Cy5-dUTP.

As described in the response to major comment 4, we decided to retract the original Fig. 2A and 2B, and replaced them with the new Fig. 2A (the requirement of Mre11 for interphase replisome disassembly), Fig. 2C (the effects when mirin was present throughout the 1st and 2nd incubations), and Fig. 2B (the same with the original 2C). However, the main conclusion of this study was not affected.

–Page 13: related to Figure 4C "CDK1-AF was able to promote mitotic entry and MRD even in the presence of mirin". Please modify this conclusion to indicate clearly that both processes occur less efficiently than in the absence of mirin.

As noted, we revised the conclusion and clearly described that both processes occur less efficiently than in the absence of mirin together with additional explanations.

–'Referee Cross-Comments'

–Taking into account the comments of the other referees, I have decided to remove my first claim that could be technically challenging.

–"1. It would be great to show that Mre11 actually processes stalled forks after the addition of the M-phase extract. Could this be revealed by primer extension by T4 DNA polymerase (with a radioactive or fluorescent dNTP) and denaturing gels as performed in Hashimoto et al., NSMB 2010? Or with another assay (DNA fibers)?"

We would like to try these experiments in a future study.

February 1, 2022

Re: Life Science Alliance manuscript #LSA-2021-01249-TR

Dr. Yoshitami Hashimoto
Tokyo University of Pharmacy and Life Sciences
School of Life Sciences
1432-1
Horinouchi
Hachioji, Tokyo 192-0392
Japan

Dear Dr. Hashimoto,

Thank you for submitting your revised manuscript entitled "Mre11 exonuclease activity promotes irreversible mitotic progression under replication stress" to Life Science Alliance. The manuscript has been seen by the original reviewers whose comments are appended below. While the reviewers continue to be overall positive about the work in terms of its suitability for Life Science Alliance, some important issues remain.

Our general policy is that papers are considered through only one revision cycle; however, given that the suggested changes are relatively minor and you may already have the data available, we are open to one additional short round of revision. Please note that I will expect to make a final decision without additional reviewer input upon resubmission.

Please submit the final revision within one month, along with a letter that includes a point by point response to the remaining reviewer comments.

To upload the revised version of your manuscript, please log in to your account: <https://lsa.msubmit.net/cgi-bin/main.plex>
You will be guided to complete the submission of your revised manuscript and to fill in all necessary information.

B. MANUSCRIPT ORGANIZATION AND FORMATTING:

Sincerely,

Reviewer #2 (Comments to the Authors (Required)):

The authors have done a great job at addressing my comments. The paper can be published without further delay.

Reviewer #3 (Comments to the Authors (Required)):

The authors answered to all the points I raised.

However, they misunderstood my second point: I was asking if the inhibition of Mre11 with Mirin in the first incubation ONLY would still suppress mitotic entry and replisome disassembly upon the addition of the M-phase extract. I think that this condition should be included in Figure 2C to better understand the effects of Mirin on mitotic entry.

Regarding the assessment of the checkpoint activation (new Figure 3D), an important control is missing which is the addition of Mirin WITHOUT M-phase extract. According to published data (Lee and Dunphy MBC 2013), Mirin should impede checkpoint activation in aphidicolin-treated extracts.

It seems also the case upon the addition of the M-phase extract: the signal of P-Chk1 is decreasing but this is not acknowledged in the text. The authors should quantify the signals of various experiments to get to a more robust conclusion.

Is the blot of Mcm2 a loading control in this figure? If so, it should be indicated in the figure legend.

Minor remarks:

page 3, "... to activate ATR-Chk1 checkpoint signaling to down-regulate CDK1/2 activities via Cdc25 inactivation, suppressing de novo initiation of DNA replication and cell cycle progression into G2/M phases (Saldivar et al, 2017). To drive these events, Cdc25 ..."

It is not clear what "these events" are referring to. I do not think that the authors refer to the suppression of both replication initiation and cell cycle progression.

Please rephrase.

page 4, results paragraph: XenOpus

26 February 2022

Dear Dr Eric Sawey,
Executive Editor
Life Science Alliance

Re: “Mr11 exonuclease activity promotes irreversible mitotic progression under replication stress” by Yoshitami Hashimoto and Hirofumi Tanaka to *Life Science Alliances*; the manuscript #LSA-2021-01249-TR.

We are grateful to you and the reviewers for your careful consideration of our revised manuscript. Following the comments by reviewer #3, we performed further experiments and modified our manuscript. We have documented our responses to the comments by reviewer #3 in the following pages.

We hope that you will consider the second revised version suitable for publication in *Life Science Alliance*.

Yours Sincerely,

Yoshitami HASHIMOTO, Ph.D.
School of Life Sciences
Tokyo University of Pharmacy and Life Sciences
1432-1 Horinouchi, Hachioji, Tokyo 192-0392, Japan
E-mail: hashimo@toyaku.ac.jp
Phone: (+81)42-676-5186
Fax: (+81)42-676-5187

Responses to the reviewers' comments

Reviewer #2 (Comments to the Authors (Required)):

–The authors have done a great job at addressing my comments. The paper can be

published without further delay.

Reviewer #3 (Comments to the Authors (Required)):

–The authors answered to all the points I raised.

–However, they misunderstood my second point: I was asking if the inhibition of Mre11 with Mirin in the first incubation ONLY would still suppress mitotic entry and replisome disassembly upon the addition of the M-phase extract. I think that this condition should be included in Figure 2C to better understand the effects of Mirin on mitotic entry.

Following the comment, we examined whether mitotic entry and replisome disassembly (MRD) were suppressed or promoted when Mre11 was inhibited only in the first incubation. To fulfill this condition, there are two ways: one is the nuclear transfer experiment in which sperm nuclei are first incubated in the mirin-treated S-phase extract, and then isolated by centrifuge and transferred to the mirin-free M-phase extract. The other is to dilute the mirin concentration in the second incubation using excess amount of egg extract. Considering the importance of preserving the integrity of nuclear envelope before mitotic induction, we avoided the nuclear transfer and performed the dilution experiment.

In the new Fig. 2C, we showed that mitotic entry and MRD was suppressed by 100 μ M mirin, but not suppressed by 20 μ M mirin in the second incubation regardless of whether Mre11 was inhibited by 100 μ M mirin in the first incubation. These results suggest that interphase Mre11 activity is not required for mitotic entry and MRD and that early mitotic Mre11 activity is necessary and sufficient for mitotic entry and MRD.

–Regarding the assessment of the checkpoint activation (new Figure 3D), an important control is missing which is the addition of Mirin WITHOUT M-phase extract. According to published data (Lee and Dunphy MBC 2013), Mirin should impede checkpoint activation in aphidicolin-treated extracts.

It seems also the case upon the addition of the M-phase extract: the signal of P-Chk1 is decreasing but this is not acknowledged in the text. The authors should quantify the

signals of various experiments to get to a more robust conclusion.

Is the blot of Mcm2 a loading control in this figure? If so, it should be indicated in the figure legend.

Following the comment, we added the control condition of mirin (+) and M-extract (-) to the original conditions and repeated this experiment three times and quantified the P-Chk1 signal intensities. We used Cut5/TopBP1 as a loading control because Cut5/TopBP1 constantly associated with chromatin even after mitotic entry, while Mcm2 gradually dissociated from chromatin after mitotic entry.

As shown in the new Fig. 2D and 2E, we observed a small but significant reduction of P-Chk1 signals after the addition of mirin without M-extract, which is consistent with the published data (Lee and Dunphy, MBC 2013) and suggests that Mre11 has a positive role for checkpoint activation. However, activated form of Chk1 did not disappear in the presence of mirin even after adding M-extract, suggesting that Mre11 may also have a role for attenuation of activated checkpoint.

-Minor remarks:

-page 3, "... to activate ATR-Chk1 checkpoint signaling to down-regulate CDK1/2 activities via Cdc25 inactivation, suppressing de novo initiation of DNA replication and cell cycle progression into G2/M phases (Saldivar et al, 2017). To drive these events, Cdc25 ..."

-It is not clear what "these events" are referring to. I do not think that the authors refer to the suppression of both replication initiation and cell cycle progression.

Please rephrase.

In the re-revised version, we deleted the expression "these events" and stated "Without replication stress, Cdc25 usually activates Cdk1/2 during G2/M phase transition to drive replication initiation and cell cycle progression by ---."

-page 4, results paragraph: XenOpus

We corrected the misspelling (*Xenpus* to *Xenopus*).

March 1, 2022

RE: Life Science Alliance Manuscript #LSA-2021-01249-TRR

Dr. Yoshitami Hashimoto
Tokyo University of Pharmacy and Life Sciences
School of Life Sciences
1432-1
Horinouchi
Hachioji, Tokyo 192-0392
Japan

Dear Dr. Hashimoto,

Thank you for submitting your revised manuscript entitled "Mre11 exonuclease activity promotes irreversible mitotic progression under replication stress". We would be happy to publish your paper in Life Science Alliance pending final revisions necessary to meet our formatting guidelines.

- please use the [10 author names, et al.] format in your references (i.e. limit the author names to the first 10)
- we encourage you to revise the figure legends for figure 1 such that the figure panels are introduced in alphabetical order
- please upload one Source Data file per main figure

A. FINAL FILES:

B. MANUSCRIPT ORGANIZATION AND FORMATTING:

Sincerely,

March 2, 2022

RE: Life Science Alliance Manuscript #LSA-2021-01249-TRRR

Dr. Yoshitami Hashimoto
Tokyo University of Pharmacy and Life Sciences
School of Life Sciences
1432-1
Horinouchi
Hachioji, Tokyo 192-0392
Japan

Dear Dr. Hashimoto,

Thank you for submitting your Research Article entitled "Mre11 exonuclease activity promotes irreversible mitotic progression under replication stress". It is a pleasure to let you know that your manuscript is now accepted for publication in Life Science Alliance. Congratulations on this interesting work.

DISTRIBUTION OF MATERIALS:

Again, congratulations on a very nice paper. I hope you found the review process to be constructive and are pleased with how the manuscript was handled editorially. We look forward to future exciting submissions from your lab.

Sincerely,
